# Development of FGF21 Mutant with Potent Cardioprotective Effects in T2D Mice via FGFR1–AMPK-Mediated Inhibition of Oxidative Stress

**DOI:** 10.3390/ijms26146577

**Published:** 2025-07-09

**Authors:** Ziying Peng, Ling Gao, Lei Zhang, Ruina Yao, Xiaoxiao Li, Long Deng, Jinxia Fan, Lei Ying, Yang Wang

**Affiliations:** Department of Pathophysiology, School of Basic Medical Sciences, Wenzhou Medical University, Wenzhou 325035, China; 18370282497@163.com (Z.P.); 15158715137@163.com (L.G.); 13207588648@163.com (L.Z.); 18735583007@163.com (R.Y.); 17582812055@163.com (X.L.); 18635431933@163.com (L.D.); 19016509691@163.com (J.F.)

**Keywords:** fibroblast growth factor 21, type 2 diabetes, diabetic cardiomyopathy, oxidative stress, adenosine 5′-monophosphate (AMP)-activated protein kinase

## Abstract

Diabetic cardiomyopathy (DCM) in type 2 diabetes (T2D) may lead to heart failure and patient death. Fibroblast growth factor 21 (FGF21) is a therapeutic candidate for treating this disease. However, one impediment to its clinical use is its weak ability to activate downstream signaling pathways. In this study, based on our in-depth understanding of the binding properties of fibroblast growth factor receptor 1c (FGFR1c) with paracrine FGF1 and endocrine FGF21, we engineered a novel FGF21 analog named FGF21^D2D3^. This was achieved by substituting the R96–V106 region of FGF21 (the binding site with the D2–D3 domain of FGFR1c) with the corresponding region from FGF1. Structural characterization and binding affinity tests showed that the analog’s capacity to bind FGFR1c was significantly enhanced compared to wild-type FGF21 (FGF21^WT^). In a T2D mouse model, we found that FGF21^D2D3^ had greater potency than FGF21^WT^ in improving hyperlipidemia and DCM. Furthermore, mechanistic studies revealed that FGF21^D2D3^ more effectively bound FGFR1, activated AMPK, inhibited oxidative stress, and ameliorated DCM. Therefore, our data indicate that FGF21^D2D3^ is a better substitute for FGF21^WT^ in treating DCM by improving dyslipidemia and directly suppressing oxidative stress via FGFR1–AMPK activation in T2D.

## 1. Introduction

Type 2 diabetes (T2D) is a global disease with a rapidly increasing incidence and imposes severe burdens on patients [1,2]. Cardiovascular disease is the most common complication and accounts for at least 50% mortality in patients with T2D [2,3]. Among these cardiovascular diseases, diabetic cardiomyopathy (DCM) is characterized by impaired cardiac structure and function [1]. Studies have shown that DCM is closely associated with metabolic stressors, including insulin resistance, chronic hyperglycemia, and dysregulated glucose and lipid metabolism, which ultimately lead to heart failure and even death [1,2,3,4,5]. However, effective therapeutics for treating DCM in T2D are still lacking [1].

As an atypical fibroblast growth factor (FGF), FGF21 is the 21st member of FGF superfamily, along with FGF15/19 and FGF23, and functions as an endocrine hormone [6,7,8]. FGF21 has been reported as a stress hormone in both physiological and pathological conditions, regulating local and systemic metabolic homeostasis and maintaining normal bodily function [8,9]. Apart from its role in stress responses, pharmacological administration of FGF21 has demonstrated various metabolic benefits in rodents, including weight reduction and alleviation of obesity, diabetes, and fatty liver disease [10,11,12,13]. In T2D monkeys, FGF21 reduces fasting plasma glucose, insulin, glucagon, and triglycerides levels [14]. Additionally, diabetic cardiomyopathy in T2D is a putative target for FGF21, based on the fact that its receptor (FGFR1) and co-receptor β-klotho (KLB), which together confer specific responsiveness to FGF21 signaling, are expressed in primary rat cardiomyocytes as well as in rodent and human hearts [15,16,17,18]. In T2D patients, circulating FGF21 levels are elevated and associated with a higher risk of cardiovascular events [19,20]. In rodents, exogenous administration of FGF21 has also shown promise in ameliorating DCM in T2D, possibly through AMPK mediated antioxidation and lipid-lowering effects, as well as sirtuin 3 axis-mediated regulation of mitochondrial integrity [17,21].

Unlike other FGFs, FGF21 is a unique metabolic hormone without mitogenic activity, indicating its potential for pharmacological application [10,22,23]. However, one impediment to its clinical use is its weak capacity to activate downstream singling pathways [24,25]. We previously reported that paracrine FGF1 has a stronger signaling capacity than endocrine FGF21, likely due to its higher binding affinity with fibroblast growth factor receptor 1c (FGFR1c) [23,25]. Based on detailed analysis of the receptor-binding characteristics of paracrine FGF1 and endocrine FGF21, we constructed an FGF21 mutant (named FGF21^D2D3^) by substituting the R96–V106 region of FGF21 with the corresponding region of FGF1, which is responsible for binding the D2–D3 domain of FGFR1. Indeed, we found that FGF21^D2D3^ exhibited stronger binding affinity for FGFR1 and more robust activation of downstream signaling pathways compared to wild type FGF21 (FGF21^WT^). We also found that FGF21^D2D3^ had a more potent cardioprotective effect than FGF21^WT^ in T2D mice, likely via FGFR1–AMPK-mediated inhibition of oxidative stress. Therefore, our data indicate that FGF21^D2D3^ is a promising therapeutic for alleviating cardiomyopathy in T2D.

## 2. Results

### 2.1. Design of FGF21 Analog with Enhanced Affinity Toward FGFR1c

Previous studies have reported that paracrine FGF1 has a stronger capacity to activate downstream signaling pathways compared to endocrine FGF21, partly due to its higher binding affinity for FGFRs, especially FGFR1c (Appendix A) [23,25]. Therefore, we compared and analyzed the crystal structures of the endocrine FGF21–FGFR1c and paracrine FGF1–FGFR1c complexes, and found that the FGFR1 binding sites consist of three parts: the D2 binding region, the D2–D3 linker region, and the D3 binding region (Figure 1A–D). Notably, the D2 and D3 binding regions between paracrine FGF1 and endocrine FGF21 are essentially the same (Figure 1B,D). However, we observed that E102-N110-Y112 of FGF1 form a more stable intramolecular hydrogen bond network with R251—a conserved amino acid in all FGFRs—located in D2–D3 domain of FGFR1c, which contributes to the strong receptor affinity of FGF1 for FGFR1c (Figure 1C,E). Although these three amino acids are conserved in FGF21 (E97-N105-Y107), the surrounding amino acid sequences in this region can lead to the loss of intramolecular hydrogen bond networks (Figure 1C,E). In particular, the hydrophilic amino acid T111 of FGF1, which is highly conserved in all paracrine FGFs, is replaced by the hydrophobic alanine in endocrine FGF19, FGF21, and FGF23 (Figure 1C,E). In addition, the surrounding amino acids near E97 in FGF21 may also affect its conformation, thereby reducing its stabilizing effect on N105 of FGF21 (Figure 1C,E). To improve FGFR1c binding affinity, we substituted the R96–V106 region of FGF21 with the corresponding L101–T111 segment of FGF1, generating a new FGF21 mutant (named FGF21^D2D3^) (Figure 1C,E). Indeed, surface plasmon resonance spectroscopy analysis confirmed that FGF21^D2D3^ exhibited higher binding affinity for FGFR1c compared to wild-type FGF21 (FGF21^WT^) (Figure 1F,G). Overall, these data indicate that FGF21^D2D3^ has stronger FGFR1c binding affinity and a greater ability to activate downstream signaling pathways than FGF21^WT^.

### 2.2. FGF21^D2D3^ Has a More Potent Lipid-Lowering Effect than FGF21^WT^ in T2D Mice

To explore the glucose- and lipid-lowering effects of FGF21^WT^ and FGF21^D2D3^ in T2D mice, we established a high-fat diet (HFD)/streptozotocin (STZ)-induced T2D mouse model. Mice were intraperitoneally injected with FGF21^WT^ (1 mg/kg body weight), FGF21^D2D3^ (1 mg/kg body weight,) or buffer control for 31 consecutive days (Figure 2A). In agreement with previous reports [26,27,28], we found that FGF21^WT^ elicited a significant glucose-lowering effect accompanied by improved insulin resistance in T2D mice (Figure 2B–D). Compared to FGF21^WT^, FGF21^D2D3^ showed a comparable glucose-lowering effect and similar enhancement in insulin sensitivity (Figure 2B–D).

We further explored the effects of FGF21^WT^ and FGF21^D2D3^ on systemic and local lipid metabolism in T2D. We found that long-term administration of FGF21^WT^ also restored dysregulated lipid metabolism in T2D mice. Serum levels of triglyceride and total cholesterol were reduced by FGF21^WT^ (Figure 3A,B). One major pathological change in the liver during T2D is local lipid metabolism dysregulation, which may contribute to systemic lipid disorders [29,30]. In the liver, FGF21^WT^ reduced the number of lipid droplets (as revealed by Oil red O staining) (Figure 3C–E), likely through activation of acetyl-CoA carboxylase (ACC) (Figure 3F). The phosphorylation and protein expression levels of ACC, which were impaired by T2D, were partially restored by FGF21^WT^. Interestingly, we found that FGF21^D2D3^ had a stronger lipid-lowering effect and more robust activation of ACC compared to FGF^WT^, both systemically and locally (Figure 3C–F). Overall, these data indicate that FGF21^D2D3^ is superior to FGF21^WT^ in regulating systemic and local lipid metabolism in T2DM mice.

### 2.3. FGF21^D2D3^ Has a More Potent Cardio-Protective Effect than FGF21^WT^ in T2D Mice

As hyperglycemia, insulin resistance, and hyperlipidemia are major contributors to DCM in T2D [1,2,3,4,5], we further explored whether FGF21 could mitigate heart damage in T2D mice. We found numerous abnormally arranged cardiac muscle cells in the hearts of T2D mice, whereas cells arrangement normalized following treatment with either FGF21^WT^ or FGF21^D2D3^ (Figure 4A). In addition, both FGF21^WT^ or FGF21^D2D3^ markedly reduced the number of inflammatory cells and the degree of myocardial fibrosis in the hearts of T2D mice (Figure 4B–E). More importantly, these beneficial effects on DCM in T2D mice were more pronounced with FGF21^D2D3^ compared to FGF21^WT^ (Figure 4A–E). Overall, these data indicate that FGF21^D2D3^ offers stronger protection against DCM in T2D mice than FGF21^WT^.

### 2.4. FGF21^D2D3^ More Potently Alleviates Cardiac Damage than FGF21^WT^ via FGFR1–AMPKa-Mediated Inhibition of Oxidative Stress in T2D Mice

Metabolic disorders in T2D initiate and exacerbate oxidative stress, a primary cause of cardiac dysfunction [1,31,32,33,34]. We next examined the oxidative status of hearts treated from HFD–STZ-induced T2D mice treated with either FGF21^WT^ or FGF21^D2D3^ by immunostaining with dihydroethidium (DHE). Immunofluorescence analysis showed significantly fewer DHE-positive cells in the hearts of T2D mouse models treated with FGF21^WT^ or FGF21^D2D3^ compared to vehicle-treated mice, with FGF21^D2D3^ showing a greater reduction than FGF21^WT^ (Figure 5A,B). In addition, the degree of cardiac lipid oxidation, measured by MDA levels, was significantly decreased, while the content of the antioxidant enzyme SOD was increased in FGF21^D2D3^-treated mice—both changes more pronounced than those observed with FGF21^WT^ treatment (Figure 5C,D).

Because FGF21^D2D3^ binds more tightly to FGFR1c than FGF21^WT^ [23,25], one possibility is that FGF21^D2D3^ activates FGFR1c to a greater extent than FGF21^WT^, contributing to its superior therapeutic effects on DCM in T2D. Indeed, we found that FGF21^D2D3^ induced a more pronounced phosphorylation level of FGFR1c in the hearts of T2D mice than FGF21^WT^ (Figure 6A). As a potential downstream signal activated by FGFR1c phosphorylation [35,36], adenosine 5′-monophosphate (AMP)-activated protein kinase (AMPK) plays a pivotal role in regulating oxidative homeostasis in cardiac cells under metabolic stress [37,38,39]. We measured AMPK phosphorylation levels in the hearts of T2D mouse models and found that FGF21^D2D3^ and FGF21^WT^ substantially restored impaired AMPK activity, with FGF21^D2D3^ producing greater effects than FGF21^WT^ (Figure 6B).

Consistent with the observed effects of FGF21^D2D3^ on DCM in vivo, we found that FGF21^D2D3^ treatment led to greater phosphorylation and activation of FGFR1 and AMPK than FGF21^WT^, both of which were suppressed by high glucose (HG) and palmitic acid (PA) in H9c2 cardiomyocytes (Figure 6C,D). Similarly, immunofluorescence analysis showed significantly fewer DHE-positive cells in H9c2 cells treated with FGF21^WT^ or FGF21^D2D3^ than in HG- and PA-treated cells (Figure 7A,B), with FGF21^D2D3^ exerting a more pronounced effect than FGF21^WT^. Moreover, FGFR1 inhibition with PD166866 [40], AMPK inhibition with Compound C [41] (Figure 7C,D), or AMPK gene silencing via siRNA significantly blocked FGF21-induced activities (Figure 8A–D).

Collectively, these data from above suggest that FGF21^D2D3^ may ameliorate cardiac pathological changes to a greater extent than FGF21^WT^ in T2D mice, likely via FGFR1–AMPKa signaling pathway-mediated inhibition of local oxidative stress in the heart.

## 3. Discussion

FGF21 is a potential therapeutic agent for DCM in T2D without mitogenic activity [10,22]. However, a major limitation of wild-type FGF21 in clinical applications is its weak activation of downstream signaling pathways [24,25]. Unlike endocrine FGF21, paracrine FGF1 more effectively activates downstream signaling pathways [23,25], likely due to its stronger and more stable binding to FGFR1c via the D2–D3 domain, mediated by the E102-N110-Y112 intramolecular hydrogen bond network. Therefore, we constructed an FGF21 mutant, named FGF21^D2D3^, by substituting its amino acids region (R96–V106 region) with the corresponding region of FGF1 (D2–D3 domain). Our results showed that FGF21^D2D3^ exhibited stronger binding affinity with FGFR1, which was further confirmed in cardiac H9c2 cells, where FGF21^D2D3^ induced greater FGFR1 phosphorylation and activation than FGF21^WT^ under high-glucose and high-lipid conditions.

In the present study, we found that both FGF21^WT^ and FGF21^D2D3^ improved hyperglycemia, insulin resistance, and hyperlipidemia in a HFD–STZ-induced T2D mouse model. Although their glucose-lowering effects were comparable, FGF21^D2D3^ showed greater potency in correcting dysregulated systemic and local lipid metabolism. These data indicate that the enhanced downstream signaling triggered by FGF21^D2D3^ is more effective in alleviating hyperlipidemia than hyperglycemia. This difference may be attributed to the fact that white adipose tissue and the liver—main organs responsible for systemic and local lipid metabolism [42]—express high levels of β-klotho, while skeletal muscle, which mediates up to ~75% of insulin-dependent glucose disposal [43,44]), does not [36,45]. Since hyperglycemia, insulin resistance, and hyperlipidemia are key contributors to the onset and progression of DCM in T2D [1,2,3], FGF21 likely treats DCM by targeting these metabolic disorders. More importantly, FGF21^D2D3^ may offer superior cardioprotective effects compared to FGF21^WT^, partially due to its greater ability to improve dysregulated lipid metabolism.

Beyond regulating metabolic disorders in T2D, FGF21 may also directly act on the heart to ameliorate DCM, as FGFR1 and KLB—the receptor and co-receptor mediating the biological functions of FGF21—are expressed in cardiac tissue [15,16,17,18]). Indeed, we found that FGF21^D2D3^ induced greater FGFR1 phosphorylation and activation than FGF21^WT^, both in the hearts of T2D mice and in H9c2 cells. This finding was further strengthened by experiments using an FGFR antagonist, which compromised most biological effects of the FGF21 mutant in vitro. Therefore, it is logical to postulate that FGF21^D2D3^ partially improves DCM through direct cardiac action.

Hyperglycemia and hyperlipidemia are common pathological features of T2D [46,47]. Increasing evidence suggests that the oxidative stress initiated by these conditions plays a major role in the occurrence and development of cardiovascular complications, including DCM [33,34,48,49,50]. We found that oxidative stress was worsened in the hearts of T2D mice and in H9c2 cardiomyocytes under high-glucose and high-lipid conditions. By contrast, this exacerbated oxidative burden was mostly abrogated by FGF21^D2D3^, which was more effective than FGF21^WT^. Furthermore, the inhibitory effect of FGF21^D2D3^ on oxidative stress was compromised by the FGFR1 antagonist, indicating that FGF21^D2D3^ alleviates DCM partially by directly inhibiting oxidative stress in the hearts of T2D mice.

AMPK is a well-studied cellular energy sensor activated by metabolic stress to maintain metabolic homeostasis, but its activity is impaired by long-lasting metabolic burdens, as seen in T2D [51]. Conversely, physiological or pharmacological activation of AMPK improves insulin sensitivity and overall metabolic health [39,51,52]. In cardiac cells, activation of AMPK can also suppress oxidative stress-induced injury [17,37]. In our study, AMPK activity was reduced in both the hearts of T2D mice and in H9c2 cells under high-glucose and high-lipid conditions, in parallel with increased ROS production. FGF21^D2D3^ more effectively increased AMPK phosphorylation levels and reduced oxidative stress than FGF21^WT^, and these effects were mostly blocked by its antagonist or siRNA. The data suggest that FGF21^D2D3^ may activate AMPK to suppress oxidative stress and thereby alleviate DCM.

## 4. Materials and Methods

### 4.1. Expression and Purification of Wild-Type FGF21 and Its Mutant

Recombinant human wild-type FGF21 (FGF21^WT^) (His29-Ser209) and its mutant (FGF21^D2D3^) were expressed using the pET28a bacterial expression vector transformed into Escherichia coli BL21 (DE3). Protein expression was induced with 1 mM isopropyl β-D-1-thiogalactopyranoside at 37 °C for 4 h, and cells were collected by centrifugation. Both FGF21 and its mutant were refolded in vitro from isolated bacterial inclusion bodies and purified using a published protocol [53].

### 4.2. Surface Plasmon Resonance (SPR) Spectroscopy

A BIA-core T200 system (GE Healthcare, Piscataway, NJ, USA) was used to analyze the real-time biomolecular interactions of FGF21^WT^ and FGF21^D2D3^ with FGFR1c ectodomain in HBS–EP buffer (10 mM HEPES-NaOH, pH 7.4, 150 mM NaCl, 3 mM EDTA, and 0.005% Surfactant P20) as previously described [54]. An amine coupling kit (GE Healthcare, Piscataway, NJ, USA) was used to immobilize ligand-binding region of FGFR1c onto the flow channels of a CM5 sensor chip. Unreacted surfaces were blocked with 1M ethanolamine-HCl (pH 8.5). As described above, control flow channels were prepared exactly the same but without the protein. Increasing concentrations of FGF21^WT^ and FGF21^D2D3^ were diluted with HBS–EP buffer and injected over the FGFR1c chip for 180 s at a flow rate of 50 μL/min; HBS–EP buffer was then flowed for 180 s to monitor dissociation at 50 μL/min. Sensor chips surfaces were regenerated by injecting either 1.5 M NaCl in 10 mM HEPES-NaOH (pH 7.5) or 2.0 M NaCl in 10 mM sodium acetate (pH 4.5). For each injection, non-specific responses from the control flow channel were subtracted from those recorded for the FGF21^WT^ and FGF21^D2D3^ flow channels. Data were processed using BIA-Evaluation software (V 3.0, GE Healthcare, Piscataway, NJ, USA), and equilibrium dissociation constants (K_d_) were calculated from fitted saturation binding curves.

### 4.3. Animals and Animal Welfare

Eight-week-old male C57BL/6J mice were purchased from Beijing Vital River Laboratory Animal Technology Co., Ltd. (Beijing, China). All animal protocols in this study were approved by the Animal Care and Use Committee of Wenzhou Medical University, China. All mice were kept in a specific pathogen-free and controlled environment (22 ± 2 °C, 55–65% humidity, 12 h light/dark cycle) and had ad libitum access to food and water. Animals were acclimatized to our laboratory environment for one week prior to formal experiments.

According to previous studies [35,55], male C57BL/6 J mice (eight-weeks-old) were fed a high-fat diet (60% fat; Dyets, Inc., Wuxi, China) for two months to initiate insulin resistance. Subsequently, mice received intraperitoneal injections with STZ (35 mg/Kg body weight in citrate buffer, pH 4.5) for consecutive four days. Simultaneously, control mice received the same volume of citrate buffer. One week post-treatment, fasting blood glucose levels were measured using a FreeStyle complete blood glucose monitor (Roche Pharmaceuticals, Branchburg, NJ, USA), mice with blood glucose levels higher than 300 mg/dL (16.7 mM) were considered T2D.

### 4.4. Functional Evaluation of FGF21^WT^ and FGF21^D2D3^ in T2D Mice

All diabetic mice were randomly divided according to their body weight and blood glucose levels, as previously described. T2D mice were interperitoneally (i.p.) injected with FGF21^WT^, FGF21^D2D3^, or buffer control for 31 consecutive days (1 mg/kg body weight). Blood samples were collected via tail snip, and glucose levels were measured every three days. Following the final dose, intraperitoneal glucose tolerance tests (IPGTTs) were conducted on T2D mice after fasting overnight (12 h); mice were i.p. injected with a dextrose solution (1.0 g/kg body weight), and blood samples were collected at designated time points for glucose measurement. The area under the curve (AUC) for IPGTTs was calculated using the trapezoid rule for glucose tolerance curves using GraphPad Prism 10 software (GraphPad Software, San Diego, CA, USA). Serum cholesterol and triglyceride levels were determined based on protocols provided by the Nanjing Jiancheng Bioengineering Institute (Nanjing, China).

### 4.5. Immunohistochemical and Immunofluorescent Staining of Mouse Tissues

Liver and heart tissues were excised from all mice and weighed. Tissues were fixed in 4% paraformaldehyde overnight and embedded in paraffin or in Tissue-Tek OCT compound (Sakura, Tokyo, Japan). After deparaffinization and rehydration, 5 μm paraffin sections were stained with hematoxylin and eosin (H&E) reagent, Oil red O solution, or picrosirius red solution using standard procedures, and examined via light microscopy (Nikon, Tokyo, Japan).

For immunofluorescent staining, heart paraffin sections (5 μm) were incubated overnight at 4 °C with a rat monoclonal anti-F4/80 antibody (FITC) (1:200) (Abcam, Cambridge, UK) and visualized with a fluorescence microscope (Zeiss, Oberkochen, Germany).

### 4.6. MDA and SOD Measurements

Mice heart tissues were homogenized in normal saline, and MDA and SOD levels were measured according to protocols provided by the Nanjing Jiancheng Bioengineering Institute, (Nanjing, China).

### 4.7. Experiments Using H9c2 Cardiomyocytes

Rat H9c2 cardiomyocytes were obtained from the National Collection of Authenticated Cell Cultures (Shanghai, China) and cultured in high-glucose DMEM (Gibco, Thermo Fisher Scientific, Waltham, MA, USA) supplemented with 10% fetal bovine serum (Wuhan Pricella Biotechnology Co., Ltd., Wuhan, China) and 1% penicillin/streptomycin (P/S) (Gibco, Thermo Fisher Scientific, Waltham, MA, USA). To measure intracellular signaling, H9c2 cells were starved in serum-free low-glucose DMEM (Gibco, Thermo Fisher Scientific, Waltham, MA, USA) for 12 h, then pretreated with PD166866 (10 μM) (Selleck, Shanghai, China), Compound C (10 μM) (Selleck, Shanghai, China), or buffer control. One hour after treatment, cells were incubated with FGF21^WT^ (500 ng/mL), FGF21^D2D3^ (500 ng/mL), or buffer control for 4 h. They were then exposed to high glucose (HG) (35 mM) plus palmitic acid (PA) (62.5 μM) for 24 h before being lysed for downstream signaling analysis via Western blot.

To knock down AMPKα expression using siRNA, cells were seeded in six-well plates and cultured to 50% confluence. Transient transfections were performed with riboFECT™ CP transfection reagent following the manufacturer’s instructions (Guangzhou RiboBio Co., Ltd., Guangzhou, China). After transfected with control (Guangzhou RiboBio Co., Ltd., Guangzhou, China) or AMPK siRNA (Guangzhou RiboBio Co., Ltd., Guangzhou, China) for 24 h, the cells were starved for another 12 h and treated as described above.

### 4.8. Western Blot Analysis

Mice livers, hearts, and H9c2 cell samples were homogenized in RIPA lysis buffer (25 mM Tris, pH 7.6; 150 mM NaCl; 1%NP-40; 1% sodium deoxycholate; 0.1% SDS) supplemented with protease and phosphatase inhibitors (Shanghai Epizyme Biomedical Technology Co., Ltd., Shanghai, China). Protein concentrations were measured using a BCA Kit (Protein Assay Kit, Shanghai Epizyme Biomedical Technology Co., Ltd., Shanghai, China). To determine protein phosphorylation and expression levels, equal quantities of soluble protein (50 μg) were separated via 10–12% SDS-PAGE and electro-transferred onto a nitrocellulose membrane. Protein blots were probed with primary antibodies against p-FGFR1 (anti-rabbit; dilution: 1:1000; cat. no. 16737; Cell Signaling Technology, Danvers, MA, USA), FGFR1 (anti-rabbit; dilution: 1:1000; cat. no. 16810; Cell Signaling Technology, Danvers, MA, USA), p-AMPK (anti-rabbit; dilution: 1:1000; cat. no. 2535; Cell Signaling Technology, Danvers, MA, USA), AMPK (anti-rabbit; dilution: 1:1000; cat. no. 5831; Cell Signaling Technology, Danvers, MA, USA), p-ACC (anti-rabbit; dilution: 1:1000; cat. no. 11818; Cell Signaling Technology, Danvers, MA, USA), ACC (anti-rabbit; dilution: 1:1000; cat. no. 3662; Cell Signaling Technology, Danvers, MA, USA), β-Tublin (anti-mouse dilution: 1:5000; cat. no. M30109; Abmart Shanghai Co., Ltd., Shanghai, China), and GAPDH (anti-rabbit; dilution: 1:5000; cat. No. 81640-5-RR; Proteintech Group, Inc., Wuhan, China). Immune-reactive bands were detected by incubation with horseradish peroxidase–conjugated goat anti-mouse (dilution: 1:5000; cat. No. HS201-01; TransGen Biotech, Beijing, China) or anti-rabbit secondary antibodies (dilution: 1:5000; cat. No. HS101-01; TransGen Biotech, Beijing, China) and visualized using enhanced chemiluminescence (ECL) reagents (Bio-Rad, Hercules, CA, USA). The optical densities of the immunoblots were analyzed using ImageJ software (version 1.42q, NIH, Bethesda, MD, USA) and normalized to the scanning signals of their respective controls. 

### 4.9. DHE Staining

Frozen heart sections (5 μm) were incubated with dihydroethidium (DHE) (Beyotime Biotechnology, Shanghai, China) (1.5 mmol/L) for 30 min and visualized by fluorescence microscopy (Nikon, Tokyo, Japan). For cell samples, treated cells were washed three times with PBS, incubated with DHE (1.5 mmol/L) for 30 min, and visualized using a fluorescence microscope (Zeiss, Oberkochen, Germany).

### 4.10. Statistical Analyses

All data were expressed as mean ± SEM. Statistical analyses were performed using GraphPad Prism 10 software (GraphPad Software, San Diego, CA, USA). Unpaired Student’s *t* tests were used to compare two groups, while one-way or two-way analysis of variance (ANOVA) with post hoc tests (Tukey or Šídák) were used for multiple comparisons. A two-tailed *p*-value less than 0.05 was considered statistically significant. N represented the number of replicates in each experiment.

## 5. Conclusions

In the present study, we found that FGF21^D2D3^ improved hyperlipidemia and directly bound to FGFR1 to activate AMPK, thereby reducing oxidative stress and ameliorating DCM in T2D. Its effects were more effective than FGF21^WT^. Our findings suggest that FGF21^D2D3^ is a more potent and effective protein than FGF21^WT^ for treating DCM through improving dyslipidemia and directly inhibiting oxidative stress via FGFR1–AMPK activation in T2D.

## Figures and Tables

**Figure 1 ijms-26-06577-f001:**
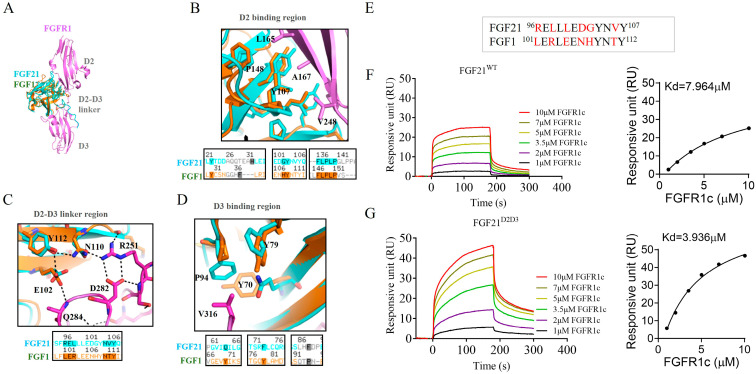
Design of a novel FGF21 analog FGF21^D2D3^ with enhanced FGFR1 binding affinity. (**A**) Overview of the structures of FGF1–FGFR1 and FGF21–FGFR1 complexes. (**B**–**D**) Structural comparisons of the D2 binding region (**B**), the D2–D3 linger region (**C**), and D3 binding region (**D**) between FGF1 and FGF21 with FGFR1. The upper panels reveal spatial differences in the binding sites of FGF1 and FGF21 with FGFR1; the lower panels show the sequence alignments of FGF1 and FGF21 in the corresponding region. (**E**) Comparison of amino acid sequences in D2–D3 linker region of FGF1 and FGF21 with FGFR1. Residues that differ between FGF1 and FGF21 are shown in red. (**F**,**G**) Representative SPR sensorgrams showing the binding interactions of FGF21^WT^ (**F**) and FGF21^D2D3^ (**G**) with the extracellular ligand-binding domain of FGFR1c (left hand side) and the saturation binding curves used to derive the equilibrium dissociation constants (K_d_) (right panels). All data are expressed as mean ± SEM.

**Figure 2 ijms-26-06577-f002:**
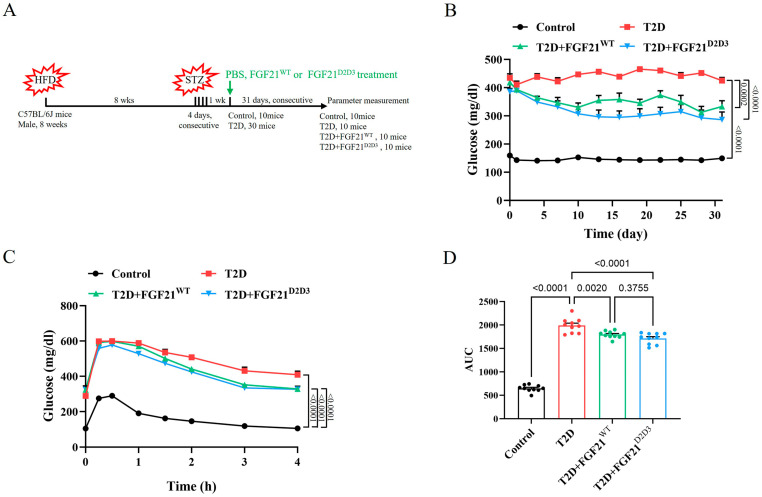
Long-term treatment with FGF21 improves hyperglycemia and insulin resistance in HFD–STZ-induced T2D mice. (**A**–**D**) HFD–STZ-induced T2D mice were i.p. administrated FGF21^WT^ (1 mg/kg body weight), FGF21^D2D3^ (1 mg/kg body weight), or buffer control for 31 consecutive days. (**A**) Schematic diagram of the experimental design. (**B**) Random fed blood glucose levels (*n* = 10). (**C**) IPGTT over the course of 4 h (*n* = 10). (**D**) Integrated AUC for changes in blood glucose levels (*n* = 10). All data are expressed as mean ± SEM.

**Figure 3 ijms-26-06577-f003:**
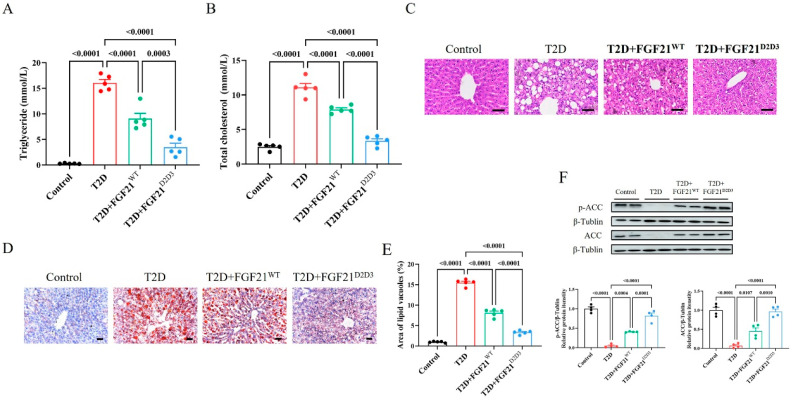
Long-term treatment with FGF21 improves systemic and local lipid metabolism in HFD–STZ-induced T2D mice. (**A**–**F**) HFD–STZ-induced T2D mice were i.p. administrated FGF21^WT^ (1 mg/kg body weight), FGF21^D2D3^ (1 mg/kg body weight), or buffer control for 31 consecutive days. (**A**) Triglyceride level (*n* = 5). (**B**) Total cholesterol level (*n* = 5). (**C**) Representative HE staining of the liver (*n* = 5); scale bar = 100 μm. (**D**,**E**) Representative Oil red O staining (**D**) of liver and its quantification (**E**) (*n* = 5); scale bar = 100 μm. (**F**) Phosphorylation and protein expression levels of ACC in the liver determined by Western blot analysis (upper panel) and quantified using ImageJ software (v1.42q) (lower panel) (*n* = 4). All data are expressed as mean ± SEM.

**Figure 4 ijms-26-06577-f004:**
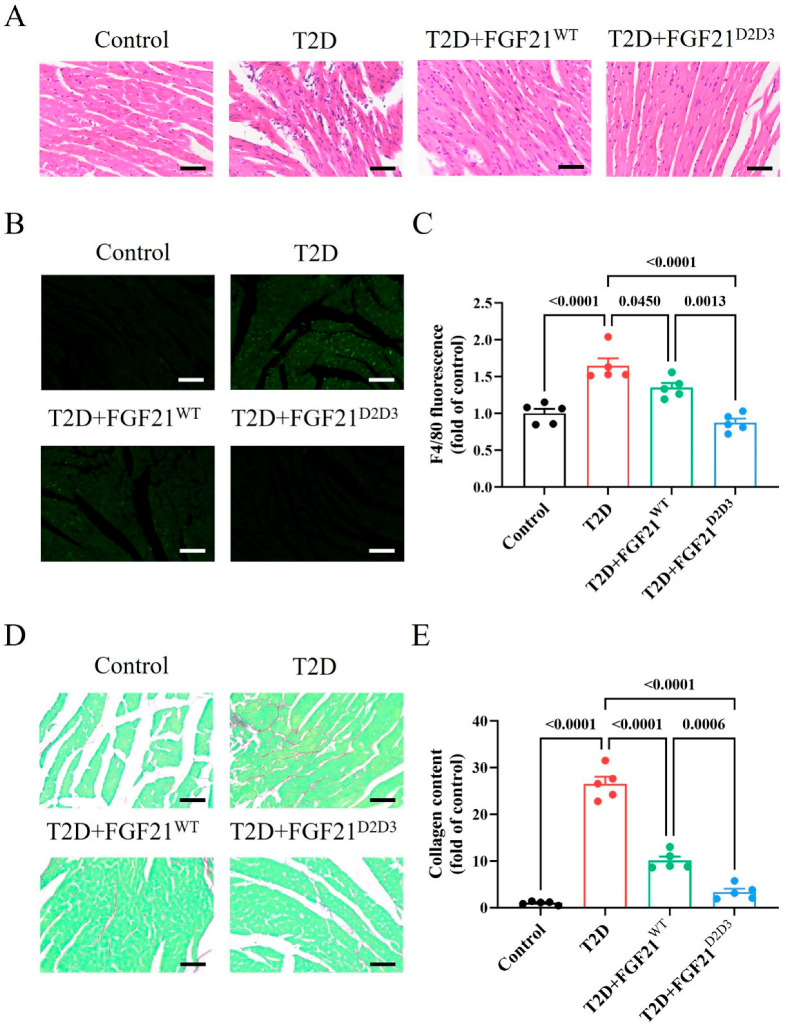
Long-term treatment of FGF21 improves DCM in HFD–STZ-induced T2D mice. (**A**–**E**) HFD–STZ-induced T2D mice were i.p. administered FGF21^WT^ (1 mg/kg body weight), FGF21^D2D3^ (1 mg/kg body weight), or buffer control for 31 consecutive days. (**A**) Representative HE staining of heart tissue (*n* = 5); scale bar = 200 μm. (**B**,**C**) Representative immunofluorescence staining of the heart with antibodies using F4/80 antibodies (**B**) and its quantification (**C**) (*n* = 5); scale bar = 200 μm. (**D**,**E**) Representative picrosirius red staining (**D**) of heart tissue and its quantification (**E**) (*n* = 5); scale bar = 200 μm. All data are expressed as mean ± SEM.

**Figure 5 ijms-26-06577-f005:**
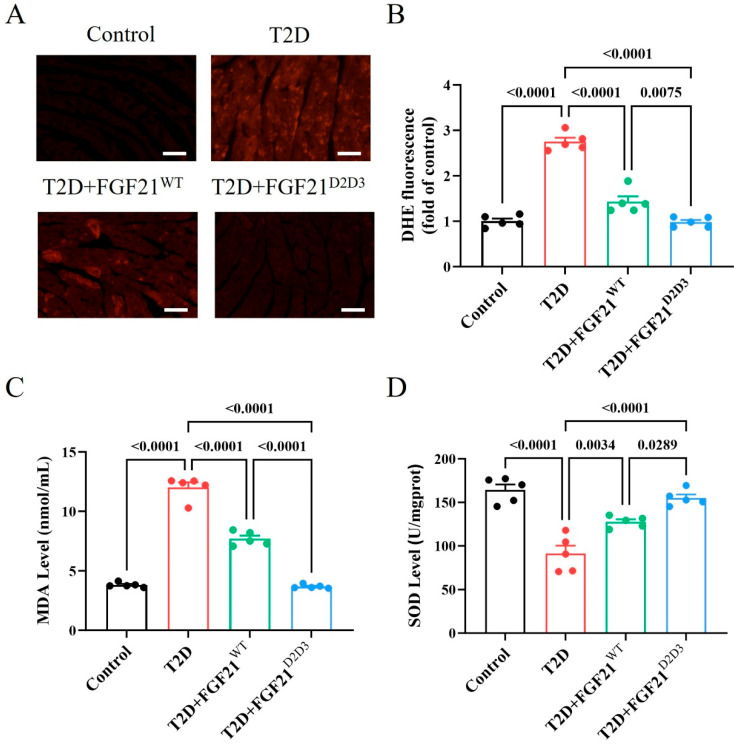
Long-term treatment with FGF21 improves oxidative stress in the hearts of HFD–STZ-induced T2D mice. (**A**–**D**) HFD–STZ-induced T2D mice were i.p. administered with FGF21^WT^ (1 mg/kg body weight), FGF21^D2D3^ (1 mg/kg body weight), or buffer control for 31 consecutive days. (**A**,**B**) Representative DHE staining of the heart (**A**) and its quantification (**B**) (*n* = 5); scale bar = 100 μm. (**C**) MDA levels in the heart. (**D**) SOD levels in the heart. All data are expressed as mean ± SEM.

**Figure 6 ijms-26-06577-f006:**
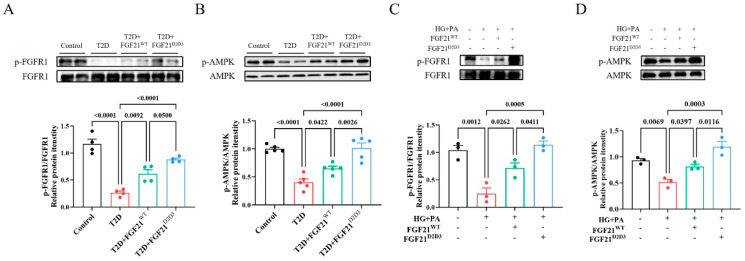
FGF21 activates FGFR1 and AMPK in the hearts of HFD−STZ-induced T2D mice and in H9c2 cardiomyocytes challenged with high glucose and palmitic acid. (**A**,**B**) HFD−STZ-induced T2D mice were i.p. administered with FGF21^WT^ (1 mg/kg body weight), FGF21^D2D3^ (1 mg/kg body weight), or buffer control for 31 consecutive days. (**A**) FGFR1 phosphorylation levels in the heart determined by Western blot analysis (upper panel) and quantified using ImageJ software (v1.42q) (lower panel) (*n* = 4). (**B**) AMPK phosphorylation levels in the heart determined by Western blot analysis (upper panel) and quantified using ImageJ software (v1.42q) (lower panel) (*n* = 5). All data are expressed as mean ± SEM. (**C**,**D**) H9c2 cells were incubated with FGF21^WT^ (500 ng/mL), FGF21^D2D3^ (500 ng/mL), or buffer control for 4 h, followed by exposure to high glucose (HG) (35 mM) and palmitic acid (PA) (62.5 μM) for 24 h. (**C**) FGFR1 phosphorylation levels determined by Western blot analysis (upper panel) and quantified using ImageJ software (v1.42q) (lower panel) (*n* = 3). (**D**) AMPK phosphorylation levels as determined by Western blot analysis (upper panel) and quantified using ImageJ software (v1.42q) (lower panel) (*n* = 3). All data are expressed as mean ± SEM.

**Figure 7 ijms-26-06577-f007:**
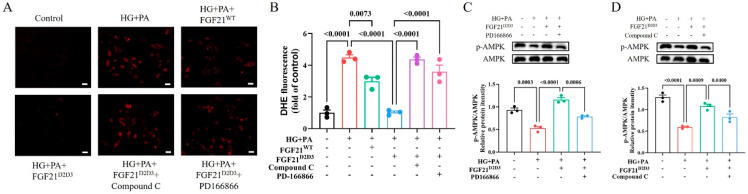
FGF21^D2D3^-mediated inhibition of oxidative stress depends on the activation of the FGFR1−AMPK signaling pathway. (**A**−**D**) H9c2 cells were pretreated with PD166866 (10 μM), Compound **C** (10 μM), or buffer control one hour prior to a 4 h incubation with FGF21^D2D3^ (500 ng/mL) or buffer control. The cells were then exposed to high glucose (HG) (35 mM) and palmitic acid (PA) (62.5 μM) for another 24 h. (**A**,**B**) Representative DHE staining of the cells (**A**) and its quantification (**B**) (*n* = 3); scale bar = 50 μm. (**C**) AMPK phosphorylation levels determined by Western blot analysis (upper panel) and quantified using ImageJ software (v1.42q) (lower panel) (*n* = 3). (**D**) AMPK phosphorylation level determined by Western blot analysis (upper panel) and quantified using ImageJ software (v1.42q) (lower panel) (*n* = 3). All data are expressed as mean ± SEM.

**Figure 8 ijms-26-06577-f008:**
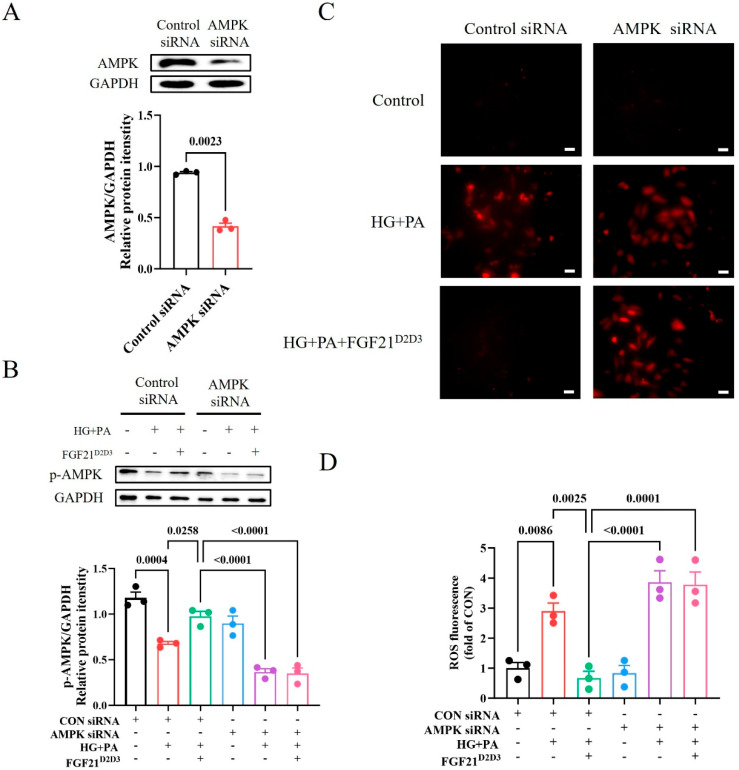
Knockdown of AMPK abrogates activation of AMPKα and inhibition of oxidative stress by FGF21^D2D3^. (**A**−**D**) H9c2 cells were transfected with control or AMPK siRNA and incubated with FGF21^D2D3^ (500 ng/mL) or buffer control for 4 h. The cells were then exposed to high glucose (HG) (35 mM) and palmitic acid (PA) (62.5 μM) for another 24 h. (**A**) AMPK protein levels determined by Western blot analysis (upper panel) and quantified using ImageJ software (v1.42q) (lower panel) (*n* = 3). (**B**) AMPK phosphorylation levels determined by Western blot analysis (upper panel) and quantified using ImageJ software (v1.42q) (lower panel) (*n* = 3). (**C**,**D**) Representative DHE staining of H9c2 cells (**C**) and its quantification (**D**) (*n* = 3); scale bar = 50 μm. All data are expressed as mean ± SEM.

## Data Availability

Data are provided within the manuscript or in Appendix A.

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
