# Peer review of "Development of FGF21 Mutant with Potent Cardioprotective Effects in T2D Mice via FGFR1–AMPK-Mediated Inhibition of Oxidative Stress"

_ijms, 2025, doi:10.3390/ijms26146577_

Round 1

Reviewer 1 Report

Comments and Suggestions for Authors

Ziying Peng et al. reported a novel engineered FGF21 analog, termed FGF21D2D3, which outperforms wild-type FGF21 in alleviating diabetic cardiomyopathy (DCM) by ameliorating dyslipidemia and suppressing oxidative stress through FGFR1-mediated AMPK activation in obese models. The authors conducted a comprehensive investigation into the downstream mechanisms of FGF21D2D3 both in vivo and in vitro. This is an interesting and meaningful study that may provide new therapeutic strategies for DCM. However, several concerns should be addressed to improve the manuscript and make it suitable for publication in the International Journal of Molecular Sciences:

Pharmacokinetics:

The in vivo metabolic profile and half-life of the FGF21D2D3 analog should be evaluated. Such pharmacokinetic data are essential for interpreting the therapeutic potential and translational relevance of this engineered molecule.

Metabolic Testing:

In Figure 2, the authors show that FGF21D2D3 improves hyperglycemia and insulin resistance in HFD-STZ-induced T2D mice. However, only a glucose tolerance test (GTT) was performed. Additional tests such as the insulin tolerance test (ITT) and pyruvate tolerance test (PTT) should be included to more comprehensively assess the systemic metabolic effects of FGF21D2D3. Moreover, it is noteworthy that this analog significantly reduces hepatic triglyceride accumulation (Figure 3), raising the question of whether it also affects body weight or adiposity in obese mice.

Cardiac Function Evaluation:

In Figure 4, FGF21D2D3 appears to improve DCM, but the current evidence is based only on histological analysis. Direct physiological measurements of cardiac function (e.g., echocardiography or pressure-volume loop analysis) would strengthen the conclusions and enhance the translational significance of the findings.

Mechanistic Validation of FGFR1 Involvement:

In Figure 7, the authors use a pharmacological inhibitor to implicate FGFR1 in the mechanism of action. However, genetic approaches (e.g., FGFR1 knockdown or knockout models) would provide more definitive evidence for the role of FGFR1 in mediating the effects of FGF21D2D3. Reliance on inhibitor treatment alone may not be sufficient due to potential off-target effects.

Author Response

Comments and Suggestions for Authors

Ziying Peng et al. reported a novel engineered FGF21 analog, termed FGF21D2D3, which outperforms wild-type FGF21 in alleviating diabetic cardiomyopathy (DCM) by ameliorating dyslipidemia and suppressing oxidative stress through FGFR1-mediated AMPK activation in obese models. The authors conducted a comprehensive investigation into the downstream mechanisms of FGF21D2D3 both in vivo and in vitro. This is an interesting and meaningful study that may provide new therapeutic strategies for DCM. However, several concerns should be addressed to improve the manuscript and make it suitable for publication in the International Journal of Molecular Sciences:

We appreciate the reviewer’s positive reception of our manuscript. The comments raised by the reviewer are constructive and very helpful for improving our manuscript.

Pharmacokinetics:

The in vivo metabolic profile and half-life of the FGF21D2D3 analog should be evaluated. Such pharmacokinetic data are essential for interpreting the therapeutic potential and translational relevance of this engineered molecule.

Response: We agree with the reviewer that evaluating pharmacokinetic data for FGF21D2D3 is important. Currently, we are unbale to exclude the possibility that the half-life of the FGF21D2D3 may be longer than wild type FGF21, which may also contribute to the enhanced metabolic benefits of FGF21D2D3. However, according to the design principle of this FGF21 mutant, the main purpose is to enhance its binding with FGFR1 and thus strengthen the signaling transduction and metabolic effects of wild type FGF21 (FGF21WT). In the study, we indeed found that FGF21D2D3 had a greater potency than FGF21WT to improve diabetic cardiomyopathy (DCM) via activating FGFR1 – AMPK signaling pathway, which should be the major reason for enhanced metabolic benefits of FGF21D2D3.

Metabolic Testing:

In Figure 2, the authors show that FGF21D2D3 improves hyperglycemia and insulin resistance in HFD-STZ-induced T2D mice. However, only a glucose tolerance test (GTT) was performed. Additional tests such as the insulin tolerance test (ITT) and pyruvate tolerance test (PTT) should be included to more comprehensively assess the systemic metabolic effects of FGF21D2D3. Moreover, it is noteworthy that this analog significantly reduces hepatic triglyceride accumulation (Figure 3), raising the question of whether it also affects body weight or adiposity in obese mice.

Response: In the study, although FGF21D2D3 improves hyperglycemia and insulin resistance in HFD-STZ-induced T2D mice, its effects are similar with those of FGF21WT. The difference of the effect of FGF21D2D3 with FGF21WT exists in their regulations on lipid disorders in T2D mice, in which FGF21D2D3 had a greater potency to correct dysregulated systemic and local lipid metabolism. This is also the reason why we focus on the regulation of FGF21D2D3 on hyperlipidemia but not hyperglycemia.

Furthermore, we measured the body weights of all mice in the study. We found that both FGF21D2D3 and FGF21WT induced conspicuous body weight lowering effects in T2D mice. However, there was no difference with the effect of FGF21D2D3 and FGF21WT .

Cardiac Function Evaluation:

In Figure 4, FGF21D2D3 appears to improve DCM, but the current evidence is based only on histological analysis. Direct physiological measurements of cardiac function (e.g., echocardiography or pressure-volume loop analysis) would strengthen the conclusions and enhance the translational significance of the findings.

 Response: Due to the lack of suitable equipment and professional technician, we are regretful that we are unable to measure cardiac function currently. However, according to previous literatures, the improvement of FGF21 on oxidative stress is closely related with the improvement of cardiac function (Leigang Jin, et al, FGF21-Sirtuin 3 Axis Confers the Protective Effects of Exercise Against Diabetic Cardiomyopathy by Governing Mitochondrial Integrity, Circulation, 2022 Nov 15;146(20):1537-1557). In the study, the worsened oxidative burden in the heart of T2D mice was mostly abrogated by FGF21D2D3, whose effect was more potent than FGF21WT. Therefore, it is reasonable to speculate that FGF21D2D3 may improve the cardiac function in T2D mice better than FGF21WT.

Mechanistic Validation of FGFR1 Involvement:

In Figure 7, the authors use a pharmacological inhibitor to implicate FGFR1 in the mechanism of action. However, genetic approaches (e.g., FGFR1 knockdown or knockout models) would provide more definitive evidence for the role of FGFR1 in mediating the effects of FGF21D2D3. Reliance on inhibitor treatment alone may not be sufficient due to potential off-target effects.

Response: In the study, we tried to knockdown FGFR1 expression in H9c2 cardiomyocyte. Due to some unknown reasons, we failed to downregulate FGFR1 expression in the cell. Therefore, we chose PD166866 as a selective FGFR1 inhibitor in the study (Panek RL, et al, In vitro biological characterization and antiangiogenic effects of PD 166866, a selective inhibitor of the FGF-1 receptor tyrosine kinase, J Pharmacol Exp Ther 1998, 286(1):569-577.).

Reviewer 2 Report

Comments and Suggestions for Authors

In this interesting work the authors focus their attention on the level of FGF21 and its role in the cardiac-vascular field in the presence of diabetes. The animal model presented is very interesting as well as the very promising results: 1-insert a figure in which the following are highlighted: number of animals and experimental procedures and times including times for sampling 2- the dysmetabolic model studied presents important characteristics that have been partly seen by the authors. Among these characteristics is a significant increase in damage also at cardiac level by the inflammasome /NLRP3 (cite and comment PMID: 28403789). this aspect has not been considered and should be considered since the fgf21/inflammasome link appears to be present in the literature (PMID: 37218012 ,PMID: 36006939 ,PMID: 32445748) 3- fgf21 has been associated with the protective role exerted by vitamin D (cite and comment PMID: 39168622 , PMID: 39544568) the authors can comment on this aspect in light of their results

Author Response

Comments and Suggestions for Authors

In this interesting work the authors focus their attention on the level of FGF21 and its role in the cardiac-vascular field in the presence of diabetes. The animal model presented is very interesting as well as the very promising results:

We appreciate the reviewer’s positive reception of our manuscript. The comments raised by the reviewer are constructive and very helpful for improving our manuscript.

1-insert a figure in which the following are highlighted: number of animals and experimental procedures and times including times for sampling

Response:As suggested by the reviewer, we have included a figure with number of animals, experimental procedures and times for sampling in the revised manuscript.

2- the dysmetabolic model studied presents important characteristics that have been partly seen by the authors. Among these characteristics is a significant increase in damage also at cardiac level by the inflammasome /NLRP3 (cite and comment PMID: 28403789). this aspect has not been considered and should be considered since the fgf21/inflammasome link appears to be present in the literature (PMID: 37218012 ,PMID: 36006939 ,PMID: 32445748)

Response:In the study, we found that inflammation in the heart of T2D mice were largely reduced by FGF21WT or FGF21D2D3, in which the beneficial effect was greater induced by FGF21D2D3 compared to FGF21WT. We agree with the reviewer that inflammasome/NLRP3 is vital for inflammation and related cardiovascular disease which can be downregulated by AMPK activation. However, we mainly focus on oxidative stress in the study which may further initiate inflammation. The relation of FGF21D2D3 and NLRP3 is worthy of studying in the future.

3- fgf21 has been associated with the protective role exerted by vitamin D (cite and comment PMID: 39168622 , PMID: 39544568) the authors can comment on this aspect in light of their results

Response:As requested by the Reviewer, we have carefully read these references. In the first reference (PMID: 39168622), the study shows that FGF23 can regulate vitamin D level to improve cardiovascular diseases including heart function. In the second reference (PMID: 39544568), it indicates that lower vitamin D level may induce lower FGF21 level. Therefore, it is unknown whether FGF21 can increase vitamin D level and thus improve diabetic cardiomyopathy in T2D. In the study we designed a FGF21 mutant (FGF21D2D3) and found it is a better substitute for FGF21WT to treat DCM. Therefore, it may be difficulty to connect FGF21D2D3 with vitamin D in the current study.

Round 2

Reviewer 1 Report

Comments and Suggestions for Authors

The authors report that, in comparison to wild-type FGF21, FGF21D2D3 analog significantly reduces hepatic lipid content but does not decrease systemic glucose metabolism by GTT or randomized fed glucose. While this finding is intriguing, I have some issues and suggestions regarding data interpretation and mechanistic understanding:

The lack of consistency between decreased hepatic lipids and systemic glucose tolerance challenges the overall metabolic effect of FGF21D2D3. We would welcome the authors to explain in detail whether the effects in liver as outlined are tissue-specific or whether more sensitive determinations (e.g., insulin tolerance tests or liver and skeletal muscle p-AKT activation) of systemic insulin sensitivity were conducted. In addition, feed blood glucose and glucose tolerance may be inappropriately low to rule out systemic metabolic changes. It would be beneficial for the authors to provide additional data such as insulin level, glucose-stimulated insulin secretion (GSIS), or hepatic gluconeogenesis by the pyruvate tolerance test (PTT) to strengthen their conclusions. Finally, authors may address possible compensatory mechanisms for maintaining systemic glucose homeostasis despite amplified hepatic function, such as increased skeletal muscle glucose disposal or varying pancreatic hormone levels.

Author Response

Reviewer 1

Comments and Suggestions for Authors

The authors report that, in comparison to wild-type FGF21, FGF21D2D3 analog significantly reduces hepatic lipid content but does not decrease systemic glucose metabolism by GTT or randomized fed glucose. While this finding is intriguing, I have some issues and suggestions regarding data interpretation and mechanistic understanding:

The lack of consistency between decreased hepatic lipids and systemic glucose tolerance challenges the overall metabolic effect of FGF21D2D3. We would welcome the authors to explain in detail whether the effects in liver as outlined are tissue-specific or whether more sensitive determinations (e.g., insulin tolerance tests or liver and skeletal muscle p-AKT activation) of systemic insulin sensitivity were conducted. In addition, feed blood glucose and glucose tolerance may be inappropriately low to rule out systemic metabolic changes. It would be beneficial for the authors to provide additional data such as insulin level, glucose-stimulated insulin secretion (GSIS), or hepatic gluconeogenesis by the pyruvate tolerance test (PTT) to strengthen their conclusions. Finally, authors may address possible compensatory mechanisms for maintaining systemic glucose homeostasis despite amplified hepatic function, such as increased skeletal muscle glucose disposal or varying pancreatic hormone levels.

We appreciate the reviewer’s positive reception of our manuscript. The comments raised by the reviewer are constructive and very helpful for improving our manuscript.

Response: We understand the concern raised by the reviewer that FGF21D2D3 significantly reduces hepatic lipid content without decreasing systemic glucose metabolism compared with FGF21WT. Based on our previous study and through searching the literature, the first reason may lay in the fact that skeletal muscle (which accounts for up to ~75% of insulin-mediated glucose disposal) (Saltiel, A. R. & Kahn, C. R. Insulin signaling and the regulation of glucose and lipid metabolism. Nature. 2001; 414, 799–806; Klip, A. & Paquet, M. R. Glucose transport and glucose transporters in muscle and their metabolic regulation. Diabetes Care. 1990; 13, 228–243) is not a putative target for FGF21 because it expresses relatively low level of β-klotho (an obligatory FGF21 co-receptor) compared with white adipose tissue (WTA) and liver (Kurosu, H. et al. Tissue-specific expression of beta Klotho and fibroblast growth factor (FGF) receptor isoforms determines metabolic activity of FGF19 and FGF21. J Biol Chem. 2007; 282, 26687–26695; Lei Ying, et al. Paracrine FGFs target skeletal muscle to exert potent anti-hyperglycemic effects. Nat Commun. 2021 Dec 14;12(1):7256). Therefore, compared to other cognate paracrine FGFs including FGF1 and FGF4 which do not require β-klotho, the glucose lowering effect of FGF21 is weak. This may also explain that why the glucose lowering effect of FGF21D2D3 is also not strong.

Secondly, it has been reported that the glucose lowering effect of FGF21 is insulin independent. Kharitonenkov A, et al. find FGF21 as a novel metabolic regulator which can induce glucose uptake in adipocyte without insulin (Kharitonenkov A, et al. FGF-21 as a novel metabolic regulator. J Clin Invest. 2005 Jun;115(6):1627-35). In insulin-resistant mice, the administration of FGF21 do not induce insulin secretion but rather lower its serum level (Jing Xu, et al. Acute glucose-lowering and insulin-sensitizing action of FGF21 in insulin-resistant mouse models--association with liver and adipose tissue effects. Am J Physiol Endocrinol Metab. 2009 Nov;297(5):E1105-14; Jimenez V, et al. FGF21 gene therapy as treatment for obesity and insulin resistance. EMBO Mol Med. 2018 Aug;10(8):e8791). In vitro, the administration of FGF21 also had no effect on insulin or glucagon secretion in either high- or low-glucose conditions (Jing Xu, et al. Acute glucose-lowering and insulin-sensitizing action of FGF21 in insulin-resistant mouse models--association with liver and adipose tissue effects. Am J Physiol Endocrinol Metab. 2009 Nov;297(5):E1105-14).

Finally, several studies have shown that FGF21 may activate AMPK in WAT and liver to exert its metabolic effects, which also decreases serum insulin level and improves glucose homeostasis independent of insulin (Mary D L Chau, et al. Fibroblast growth factor 21 regulates energy metabolism by activating the AMPK-SIRT1-PGC-1alpha pathway. Proc Natl Acad Sci USA. 2010 Jul 13;107(28):12553-8; Yu Gao, et al. Exercise and dietary intervention ameliorate high-fat diet-induced NAFLD and liver aging by inducing lipophagy. Redox Biol. 2020 Sep:36:101635; Robert W Myers, et al. Systemic pan-AMPK activator MK-8722 improves glucose homeostasis but induces cardiac hypertrophy. Science. 2017 Aug 4;357(6350):507-511). FGF21 targets WAT and liver to exert its beneficial effects including lowering of plasma insulin, FFA and triglycerides, weight loss, increased energy expenditure and reduced hepatic steatosis (Nies VJ, et al. Fibroblast Growth Factor Signaling in Metabolic Regulation. Front Endocrinol (Lausanne). 2016 Jan 19:6:193). Because WAT highly expresses FGFR1 and β-klotho, FGF21D2D3 (which has a higher binding affinity with FGFR1) may have stronger metabolic effects on WAT compared with FGF21WT. Although the liver has low level of FGFR1 expression, it expresses high level of β-klotho. Therefore, the metabolic effect of FGF21D2D3 in the liver may be also stronger than FGF21WT (Nies VJ, et al. Fibroblast Growth Factor Signaling in Metabolic Regulation. Front Endocrinol (Lausanne). 2016 Jan 19:6:193; Kurosu, H. et al. Tissue-specific expression of beta Klotho and fibroblast growth factor (FGF) receptor isoforms determines metabolic activity of FGF19 and FGF21. J Biol Chem. 2007; 282, 26687–26695). Therefore, it is logical to postulate that FGF21D2D3 has more potent lipid lowering and hepatic steatosis alleviating effects than FGF21WT in T2D mice in the study.

Based on above facts, it may explain why FGF21D2D3 has more significant effect on reducing hepatic lipid content rather than decreasing systemic glucose metabolism compared with FGF21WT. The explanation has been included in the revised manuscript and marked in red (line 266 – line 270). 

Reviewer 2 Report

Comments and Suggestions for Authors

I agree new version.

Author Response

Reviewer 2

Comments and Suggestions for Authors

I agree new version.

Response: We thank the reviewer for the positive recognition of our revised manuscript.

Round 3

Reviewer 1 Report

Comments and Suggestions for Authors

The authors have  addressed the previous concerns by providing appropriate supporting evidence. I have no further concerns.